# Chinese Traditional Pear Paste: Physicochemical Properties, Antioxidant Activities and Quality Evaluation

**DOI:** 10.3390/foods12010187

**Published:** 2023-01-01

**Authors:** Yunxiao Feng, Hong Cheng, Yudou Cheng, Jiangli Zhao, Jingang He, Nan Li, Jinxiao Wang, Junfeng Guan

**Affiliations:** 1Institute of Biotechnology and Food Science, Hebei Academy of Agricultural and Forestry Sciences, Shijiazhuang 050051, China; 2Plant Genetic Engineering Center of Hebei Province, Shijiazhuang 050051, China

**Keywords:** pear paste, physicochemical properties, antioxidant activities, factor analysis, cluster analysis

## Abstract

As a traditional folk medicine, pear paste has important nutritional and health effects. The physicochemical properties and antioxidant activities of pear pastes prepared from 23 different cultivars were investigated, including color parameters ( *L**, *a**, *b** and *h*°), transmittance, pH, titratable acidity (TA), soluble sugar content, total phenolics content (TPC), total flavonoids content (TFC), DPPH and ^•^OH radical scavenging activity (RSA), and ferric reducing antioxidant power (FRAP). It was demonstrated that the physicochemical properties and antioxidant activities of pear pastes from various cultivars differed significantly. Pear cultivars of “Mantianhong”, “Xiangshui” and “Anli” possessing higher TPC and TFC exhibited excellent antioxidant activity determined by DPPH RSA, ^•^OH RSA and FRAP, while the lowest TPC and TFC was observed for the cultivars of "Xueqing", "Nansui", "Hongxiangsu", and “Xinli No. 7”, which also demonstrated the poor antioxidant activity. Multivariate analyses, including factor and cluster analysis, were used for the quality evaluation and separation of pear pastes based on their physicochemical and antioxidant properties. Factor analysis reduced the above thirteen parameters to final four effective ones, i.e. DPPH RSA, color *b**, FRAP and TA, and subsequently these four parameters were used to construct the comprehensive evaluation prediction model for evaluating the quality of pear pastes. The pear pastes could be separated into three clusters and differentiated for the diverse of pear cultivars via cluster analysis. Consistently, “Mantianhong”, “Xiangshui” and “Anli” pear with higher quality clustered into one group, in contrast, "Xueqing", "Nansui", "Hongxiangsu", and “Xinli No. 7” with lower quality clustered into the other group. It provided a theoretical method to evaluate the quality of pear paste and may help the fruit processing industry select the more suitable pear cultivars for pear paste making.

## 1. Introduction

Pear is one of the main fruits in China with its cultivation area and production ranking the first in the world. The pear fruits have been proved to be rich in different kinds of bioactive polyphenol compounds, such as hydroxybenzoic acids, hydroxycinnamic acids, flavonoids and anthocyanins, varying with the cultivars [1,2,3]. Polyphenols have been proved to provide high nutritional and healthy benefits for humans with their antioxidant, anti-glycation and anti-inflammatory biological activities [4,5,6], and also play an important role in anti-aging, anti-cancer, and prevention of cardiovascular and neurological diseases [7,8,9].

Postharvest fruit processing can both reduce the resource waste and also add commodity value to them. Researches for pear processing mainly focus on pear juice [10,11,12,13], pear wine [14,15,16,17], and pear vinegar [18,19,20], unfortunately, there are few studies on pear paste. As a traditional folk medicine, pear paste has important nutrition function and has been proved to be beneficial for the colds and coughs of humans according to the ancient Chinese medicine book “Compendium of Materia Medica (Ben Cao Gang Mu)”, which recorded that pear paste could invigorate the body and relieve thirst, resolve phlegm and moisten the lung, alleviate cough and asthma, and relieve diarrhea and diuresis [21,22,23]. Modern studies have also proved pear paste could enhance the antioxidant capacity of cell in the rats [24]. However, knowledge about its physicochemical properties and antioxidant activities is very limited. Since nutrients of pear fruits may vary with their cultivars, the quality of pear paste prepared from different cultivars may also differ depending on cultivars. Therefore, it is necessary to investigate the quality of pear paste prepared from different cultivars and further establish the corresponding quality evaluation method.

Multivariate analyses are extensively used for quality evaluation, differentiation and classification of food processed products [25,26,27]. Among them, factor analysis is the most common method to identify potential factors that play the dominating role in the observed results, which combines multiple factors into a few factors, but could still reconstruct the correlation between the original variations and the factors [27]. With the initial assumption that the nearness of samples in the p-space defined by the variables could reflect the similarity of their properties, cluster analysis characterizes the proximity of samples (objects) to each other by measuring the distance or similarity between them [26,27]. Recently, multivariate analyses, such as factor and cluster analysis, were used to evaluate the quality of fruits, vegetables and their processed products, such as tomatoes [28], cherries [29], apple juice [30,31], peach puffed crisps [32], pears and pear juice [33,34,35], and so on. Up to now, the evaluation and differentiation methods for pear paste using multivariate analyses have not been reported.

The main aim of the present study was to evaluate and differentiate the quality of pear pastes prepared from 23 different cultivars based on their physicochemical properties and antioxidant activities. The current study will provide a better understanding for the nutritional and antioxidant properties of pear paste, and the established evaluation method could assist the fruit processing industry to select the more suitable pear cultivars for pear paste making.

## 2. Materials and Methods

### 2.1. Chemicals and Reagents 

Sodium hydroxide was purchased from Tianjin Continental Chemical Reagent Factory; potassium sodium tartrate was purchased from Bodi Chemical Co., Ltd. (Tianjin, China); sodium carbonate, gallic acid and glucose anhydrous were purchased from Bioengineering Co., Ltd. (Shanghai, China); potassium ferricyanide, copper sulfate, ferrous sulfate were purchased from Yongda Chemical Reagent Co., Ltd. (Tianjin, China); potassium ferricyanide was purchased from Beichen Huayue Chemical Reagent Co., Ltd. (Tianjin, China); Folin—Ciocalteu’s phenol reagent was purchased from Biochemical Technology Co., Ltd. (Beijing, China); aluminum nitrate was purchased from West Asia Chemical Industry Co., Ltd. (Linyi, China); salicylic acid and rutin were purchased from BBI Life Sciences Co., Ltd. (Shanghai, China); 1,1−Diphenyl−2−trinitrophenylhydrazine (DPPH) was purchased from Yuanye Biotechnology Co., Ltd. (Shanghai, China); Hydrogen peroxide, sodium nitrite, trichloroacetic acid and methylene blue were purchased from Damao Chemical Reagent Factory (Tianjin, China); ferric chloride was purchased from Solexpo Technology Co., Ltd. (Beijing, China); anthrone was purchased from Comio Company (Shanghai, China); sulfuric acid was purchased from Reagent Factory (Shijiazhuang, China).

### 2.2. Fruit Materials and Sample Preparation 

Twenty-three cultivars of pear fruits bought from the local fruit wholesale market in Shijiazhuang were included in the study, see Table 1. The fruit samples were carefully selected for fruit size, absence of external and internal damage and without pests and diseases, washed and sliced into small pieces with the core and stem removed, and then crushed into juices with a juice presser (HU24FR3L, Hurom, Korea). After squeezing the juice and passing through 100 mesh sieve, the filtered pear juice was boiled with electric pottery stove (LC−EA3S, Guangdong Shunde Zhongchen Electric Co., Foshan, China) to make pear paste. The starting power of electric pottery stove was first set at 500 W, followed by gradient change of electric pottery stove power to 300 W and 200 W when the soluble solid content (SSC) of the concentrated pear juice reached 30 and 50 °Brix, respectively, until the SSC reached 70 °Brix, ending the boiling, and finally the concentrated pear juice become highly sticky pear paste. In order to detemine the physicochemical properties and antioxidant activities of pear paste, the original pastes have to be diluted until the SSC being 10 °Brix, since the pear paste was too sticky, and the obtained dilutions were used next.

### 2.3. Measurement of Pear Firmness, Soluble Solid Content (SSC), pH, Titratable Acidity (TA), Yields of Juice and Paste

The firmness of the pear was monitored by hardometer (Top Instrument Co., Hangzhou, China). SSC (°Brix) of pear juice was measured by a portable °Brix meter (Atago Co., Ltd., Tokyo, Japan). The pH of pear juice was monitored by a pH meter (ST3100, OHAUS (Changzhou) Instruments Co., China). TA of pear juice was determined by acid-base titration with slight modification according to the previous method [36], aliquots of 5 mL of pear juice placed into a 100 mL conical flask and titrated with standardized 0.01 M NaOH until phenolphthalein end point (pH 8.2 ± 0.1), and then the volume of NaOH was converted to g per 100 g of malic acid. The juice yield was calculated by dividing the weight of juice collected (kg) by the weight of pear fruit sample (kg), and the pear paste yield was calculated by dividing the weight of pear paste collected (kg) by the weight of pear fruit sample (kg).

### 2.4. Measurement of Color and Transmittance for the Diluted Pear Paste

A colour guide system (CR−400, Konica Minolta Co., Ltd., Osaka, Japan) with Illuminant D_65_ and 10° observer angle was used to measure the color (*L**, *a** and *b**) of the diluted pear paste using a colorless and transparent petri dish filled with 20 mL diluted pear paste. *L** indicated black (*L** = 0) to white (*L** = 100) component, *a** indicated green (−) to red (+) component, and *b** indicated blue (−) to yellow (+) component. The Hue angle value showed the variation of color which could be calculated by the formula: *h*° = arctan (*b**/*a**) with the variation between 0° (purple−red) and 180° (green), and the medium h = 90° is for yellow. The transmittance (T) of the diluted pear paste was determined at 625 nm using a 1240 UV−vis spectrophotometer (Shimadzu Co., Ltd., Kyoto, Japan).

### 2.5. Measurement of TA, pH and Soluble Sugar Content for the Diluted Pear Paste

TA and pH of the diluted pear paste were determined the same as the above pear juice. Soluble sugar content was analyzed by anthrone colorimetry according to a previous study [37], and glucose with concentration ranging between 0.02 mg/mL to 0.1 mg/mL was used as a standard curve. 

### 2.6. Determination of Total Phenolics and Flavonoids Content for the Diluted Pear Paste

The total phenolics content (TPC) was determined according to the previous method reported by Jiang [13]. Briefly, 0.1 mL of the diluted sample in distilled water was mixed with 0.5 mL of 0.5 M Folin−Ciocalteu’s reagent, 4.4 mL of distilled water and 1.0 mL of 7% sodium carbonate solution. The mixture was kept in dark for 2 hours at constant temperature of 30 °C in order to allow the reaction finish, and the absorbance was measured at 760 nm with a 1240 UV—vis spectrophotometer (Shimadzu Co., Ltd., Kyoto, Japan). The concentration of gallic acid ranging between 0.01 mg/mL to 0.05 mg/mL was used as a standard curve. Meanwhile, 0.1 mL of distilled water was used as blank control, and the TPC was calculated using the standard curve and expressed as milligram of gallic acid equivalent per 100 mL of the diluted pear paste.

The total flavonoids content (TFC) was measured according to the method described by Jiang [13]. Briefly, 0.2 mL of the diluted pear paste sample in distilled water was mixed with 0.3 mL of 5% NaNO_2_ and 0.8 mL distilled water, and reacted at room temperature for 5 min. Afterwards, 0.3 mL of 10% AlCl_3_·6H_2_O was added and the solutions were continued incubated for 5 min at room temperature before the addition of 2 mL of 1 M NaOH. The absorbance at 510 nm was measured after 15 min of incubation with a 1240 UV−vis spectrophotometer (Shimadzu Co., Ltd., Kyoto, Japan). The concentration of rutin ranging between 0.02 mg/mL to 0.1 mg/mL was used as a standard curve. Meanwhile, 0.1 mL of distilled water was used as blank control, and the TFC was calculated using a standard curve and expressed as milligram of rutin equivalent per 100 mL of the diluted pear paste.

### 2.7. Determination of DPPH Radical Scavenging Activity (DPPH RSA) for the Diluted Pear Paste

The DPPH RSA of the diluted pear paste sample was determined using the method described by Jiang with slight modification [13]. Briefly, 0.05 mL of diluted sample was mixed with 3.9 mL of 60 μM DPPH. After incubating the solution at room temperature in the dark for 30 min, the absorbance of the solution was measured at 517 nm. Meanwhile, 0.05 mL of 80% ethanol with 3.9 mL of 60 μM DPPH was used as blank control, The percentage of DPPH RSA was calculated according to the equation below:$$\%\mathrm{DPPH}\;\mathrm{RSA}=\frac{A_{control}-A_{sample}}{A_{control}}\;\times \;100$$

### 2.8. Determination of ^•^OH Radical Scavenging Activity (^•^OH RSA) for the Diluted Pear Paste

The ^•^OH RSA of the diluted pear paste was determined using the previous reported method [38,39]. Briefly, 0.2 mL of diluted sample was mixed with 1 mL of 9 mM salicylic acid ethanol solution, 1 mL of 9 mM FeSO_4_ solution and 0.8 mL of distilled water, and finally 1 mL of 8.8 mM H_2_O_2_ solution was added to initiate the reaction. Simultaneously, the distilled water was used as a blank control. After the reaction at 37 °C for 30 min, both the sample solution and the blank control were centrifuged at 11,000 *g* for 6 min, and the absorbance of the supernatants of the diluted sample (*A_X_*) and the blank control (*A_0_*) was measured at 510 nm. Considering the different absorbance values of the diluted pear paste solution at 510 nm, 1 mL of 9 mmol/L FeSO_4_ solution, 1 mL of 9 mmol/L salicylic acid ethanol solution, 0.2 mL of diluted sample and 1.8 mL of distilled water was measured as the background absorbance values (*A_X0_*). The percentage of ^•^OH RSA was calculated according to the equation below:(1)$$\%\cdot \mathrm{OH}\;\mathrm{RSA}=\frac{A_{0}-\left(A_{X}-A_{X0}\right)}{A_{0}}\;\times \;100$$

### 2.9. Determination of Ferric Reducing Antioxidant Power (FRAP) Assay for the Diluted Pear Paste

The FRAP of diluted pear paste was performed according to the method of Jiang [13] with slight modification. Briefly, 0.5 mL of diluted sample was mixed with 2 mL of PBS (pH~6.6) and 2 mL of 1% potassium ferricyanide and kept in water bath at 50 °C for 20 min. Then, 2 mL of 10% trichloroacetic acid aqueous solution was added and centrifuged for 10 min at 3000× *g*. Subsequently, 0.5mL supernatant was taken and mixed well with 0.4 mL of 0.1% ferric trichloride aqueous solution and 2 mL of distilled water and kept in dark for 30 min. The absorbance of the solution was measured at 700 nm and the reducing power of pear paste was expressed as absorbance units OD_700_.

### 2.10. Statistical Analysis

All samples were analyzed in triplicate and the results were expressed as the mean ± standard deviation. Spearman correlation analysis between the mean physicochemical properties and antioxidant activities of pear pastes prepared from 23 cultivars, Duncan’s test for significance of difference, factor analysis and cluster analysis were all performed by Statistical Product and Service Solutions software SPSS 18 (IBM SPSS Statistics, Inc., Chicago, IL, United States) with the significant level set at *p* < 0.05.

## 3. Results and Discussion

### 3.1. Physicochemical Properties of Different Pear Cultivars

The single fruit weight and firmness of pear, TA, SSC and pH of pear juice, and juice and paste yields of 23 pear cultivars were seen in Table 1. The juice yields ranged between 69.57% to 88.09% with the two highest pear cultivars for “Xinli No. 7” (88.09%) and “Mantianhong” (86.45%), and the pear paste yields ranged between 8.17% to 13.16% with the two highest pear cultivars for “Shuihongxiao” (13.16%) and “xueqing” (12.73%).

### 3.2. Color Parameters and Transmittance of Diluted Pear Paste

Color is an important quality property of pear paste because it can determine its acceptability by the consumers. The color of pear paste prepared in the current study differed slightly depending on the cultivars which could be fully reflected by color parameters (*L**, *a**, *b** and *h*⸰) shown in Table 2. Among them, color *a** is corresponded to the red-green contribution, and color *b** is corresponded to the yellow-blue contribution. All samples had a positive or slightly negative *a** value (−1.16 to 4.91) which demonstrated that the samples were in red place, while all samples had positive *b** values (5.53 to 14.64) meaning that all the samples were yellow. Furthermore, the *h*⸰ value of all samples ranged between 58.91 to 97.19, indicating the pear paste showed the overlapping color of red and yellow. Briefly, all the color parameters suggested the color of pear paste were of different level of reddish brown color. The transmittance of pear pastes at 625 nm ranging from 31.10% to 90.95% could reflect the clarification degree of the sample.

### 3.3. pH, TA and Soluble Sugar of Diluted Pear paste

The pH, TA and soluble sugar were analyzed from the processed pear pastes prepared from 23 different cultivars (Table 3). pH could be used to indicate the sourness of fruit juices, wines and other processed products in the quality evaluation [1,40]. The lowest values were found in “Anli” pear (pH 3.27) and “Xiangshui” pear (pH 3.40), while the two highest values were found in “Nansui” pear (pH 5.36) and “Hongxiangsu” pear (pH 5.31). Pear paste made from different cultivars had lower pH, indicating that pear cultivar was sourer than the other studied cultivars. “Anli” pear rich in acid with the lowest pH was indeed sour, which was also called sour pear. Consistently, the highest TA was also found in “Anli” pear (0.652%), and in contrast, “Hongxiangsu” pear (0.068%) and “Nansui” pear (0.088%) had the first two lower TA, respectively. The soluble sugar content of the studied pear paste ranged between 59.47 to 84.63 mg/g.

### 3.4. Total Phenolics and Flavonoids Content of Diluted Pear Paste

The results of total phenolics and flavonoids content (TPC and TFC) were shown in Table 3, which exhibited a very obvious difference related to pear cultivars. The TPC of pear pastes ranged between 10.52 to 51.98 mg/100 mL, with the two highest pear cultivars to be “Mantianhong” pear (51.98 mg/100 mL) and “Xiangshui” pear (43.22 mg/100 mL) and the two lowest to be “Nansui” pear (10.52 mg/100 mL) and “Xueqing” pear (11.30 mg/100 mL). Like TPC, the highest level of TFC was also found in “Mantianhong” pear (42.28 mg/100 mL) and “Xiangshui” pear (20.44 mg/100 mL), so was the lowest level as “Nansui” pear (2.74 mg/100 mL) and “Xueqing” pear (3.13 mg/100 mL). According to the previous reports, phenolic profiles of pear were complex and the content of individual phenolic compounds also differed significantly in different pear cultivars [1,2,3]. The total polyphenolic content ranged differently for five typical *pyrus* species including ten pear cultivars [2]. Additionally, the contents and types of phenolic compounds in different European pear cultivars also differed significantly [1,3]. It may due to the discrepancies of the types and contents of phenolic compounds in different pear cultivars that led to the differences in TPC and TFC of pear paste. 

### 3.5. Antioxidant Activities of Diluted Pear Paste

The antioxidant activity of pear paste, including DPPH and ^•^OH radical scavenging activity (DPPH RSA and ^•^OH RSA) and ferric reducing antioxidant power (FRAP) (Table 4), are related to their ability to scavenge free radicals. The DPPH RSA ranged from 1.50% to 78.08%, with the top three be “Xiangshui” pear (78.08%), “Mantianhong” pear (69.31%) and “Anli” pear (65.84%), while the lowest three were “Nansui” pear (1.50%), “Hongxiangsu” pear (2.53%) and “Xinli No. 7” pear (6.34%). Results of ^•^OH RSA ranged between 57.02% to 96.16%, in which the first three were “Anli” pear (96.16%), “Mantianhong” pear (92.81%) and “Zaomisuan” pear (92.20%), and the last three were “Hongxiangsu” pear (57.02%), “Xinli No. 7” pear (59.72%) and “Nansui” pear (62.49%). Pear paste with the highest three FRAP were the cultivars “Mantianhong” (2.05), Mansoo (0.50) and “Xiangshui” (0.48), While the lowest three were “Xinli No. 7” pear (0.04), “Xuehua” pear (0.05) and “Xueqing” pear (0.09). Although the antioxidant evaluation methods were different, the antioxidant activities of pear paste made from “Mantianhong” pear, “Xiangshui” pear and “Anli” pear always located at the top, conversely, “Xueqing” pear, “Xinli No. 7” pear, “Hongxiangsu” pear and “Nansui” pear always seemed at the bottom of the list.

It was confirmed that the phenolics and flavonoids played a very important role in antioxidant activities, which might ascribed to hydrogen atom transfer ability of their phenolic hydroxyl groups and the stability of the formed phenoxy radicals [41]. As different pear cultivars contained different phenolics and flavonoids profile, thus the antioxidant activities of pear pastes differed significantly. Indeed, the antioxidant activities of pear pastes were found to be significantly correlated with the TPC and TFC in pear paste in our study, which was also reported in other studies [11,42]. Figure 1 showed the heat map of Spearman correlation coefficient for the physicochemical properties and antioxidant activities of 23 pear pastes. With the correlation significant level setting at *p* < 0.05, highly significant positive correlations were observed between the TPC (*r* = 0.88) and TFC (*r* = 0.86) with DPPH RSA, and highly significant positive correlations were also observed between the TPC (*r* = 0.76) and TFC (*r* = 0.71) with ^•^OH RSA. The moderate significant correlations were observed between the TPC (*r* = 0.47) with FRAP, however exception was observed for no significant correlations between the TPC (*r* = 0.39) with FRAP. Furthermore, we also found a highly significant correlation between the pH and TA of pear pastes with their antioxidant activities determined by DPPH RSA and ^•^OH RSA, shown in Figure 1, which suggested that organic acids were also closely related to the antioxidant activities of pear pastes. It has already been confirmed that not only polyphenols but also organic acids have good antioxidant activities and could be the good dietary source of antioxidants. For example, phenolic compounds and organic acids might be responsible for the antioxidant activities of fruit vinegars and *Camellia oleifera* cake according to the previous reports [43,44]. In addition, organic acids showed a synergistic effect with *α*-terpinene in DPPH scavenging activity [45].

### 3.6. Evaluation Method of the Quality of Diluted Pear Paste 

The physicochemical and antioxidant properties of each pear paste sample were performed by multivariate analyses, including factor and cluster analysis, in order to construct the comprehensive evaluation prediction model for the quality of pear pastes and to separate the pear pastes into different groups. 

Before the application of factor analysis, it was necessary to carry out the Kaiser−Meyer-Olkin (KMO) and Bartlett ball type test. KMO test was used to test the correlation between variables, if the KMO test was below 0.5, the application of factor analysis was not suitable on this occasion. While Bartlett ball type test was used to test whether the correlation matrix was a unit matrix, and the use of factor analysis should be performed carefully if the test *p* > 0.05 [27]. Based on our current research, KMO test was 0.679 greater than 0.5 and *p* < 0.001 in Bartlett’s test was less than 0.05, both indicating the data was suitable for factor analysis. Factor analysis was firstly used to seek the principal factor affecting the quality of pear pastes from 23 cultivars. A Varimax-rotation was performed in order to ensure that the abstracted principal factors were uncorrelated and facilitated the interpretation of the results. By taking eigenvalues of greater than 0.7, four principal factors (F1, F2, F3 and F4) were extracted. Table 5 showed the loadings, eigenvalues, percent of variance, cumulative variance and principal factor weight obtained from the factor analysis. The four principal factors explaining 42.4%, 25.1%, 14.6% and 8.4% of the total variance, respectively, with the cumulative variance to be 90.5%, could contain all sufficient information. According to the loading values and only taking those absolute values *p* ≥ 0.7 into consideration, pH, TA, DPPH RSA and ^•^OH RSA were the main variables correlating to F1. TFC and FRAP were correlated to F2. Color *b** was correlated to F3 and soluble sugar was the main variable correlating to F4. Then the score of the comprehensive factor (F) for all the pear paste samples could be calculated according to the score of each principal factor of pear paste sample and the corresponding principal factor weight, as the equation be: F = 0.469F1 + 0.277F2 + 0.162F3 + 0.093F4(2)

Based on this, pear pastes prepared from 23 cultivars could be ranked by the score of the comprehensive factor F (Table 6), demonstrating that the higher the score of F, the better the quality of pear paste. Among them, the top three were "Mantianhong" pear, "Xiangshui" pear and "Anli" pear, while the last three were "Xinli No. 7" pear, "Hongxiangsu" pear and "Nansui" pear, in good agreement with the the top three and bottom three of the results of the antioxidant activities.

The comprehensive evaluation prediction model for the quality of pear paste was afterwards constructed according to the multiple linear and stepwise regression, in which the above score of F was taken as the dependent variable Y, and the eight physicochemical parameters (pH, TA, DPPH RSA, ^•^OH RSA, TFC, FRAP, *b** and total soluble sugar) correlated to the four principal factors were the independent variable X. And finally, the prediction model containing only four effective parameters was obtained and expressed as:F′ = −2.084 + 0.015DPPH RSA + 0.114*b** + 0.357FRAP + 1.408TA(3)
with *r* = 0.994, *p* < 0.001. In which, F′ was the comprehensive evaluation prediction factor. The Variance inflation Factor (VIF) for the four effective parameters DPPH RSA, b*, FRAP and TA were 3.600, 1.164, 1.259, and 3.492 respectively, indicating the less multi−collinearity and no overfitting in this linear equation. The partial correlation coefficient was 0.958, 0.974, 0.912, and 0.869 for the four effective parameters separately, suggesting that the larger the absolute value, the greater its role in the comprehensive evaluation prediction model for the quality of pear paste. According to the equation, the scores of F′ (Table 6) could be calculated if we substitute the values of four effective parameters into the Equation (2). With F′ and F were highly significant positive correlation by correlation analysis, we got the comprehensive evaluation prediction model with only four effective parameters, which could evaluate the quality of pear paste.

Cluster analysis was then performed in order to test the similarity and search for groupings among the different pear paste samples based on the calculated score of F′. Three clusters were found at a similarity level of 3 using Euclidean Distance (Figure 2). The first cluster included "Xiangshui" pear, "Mantianhong" pear, and "Anli" pear, with their F′ score ranging from 0.777 to 1.154, exactly to be the top three with the highest F score. The second was made up of the medium quality of pear paste, including pear cultivars for Red pear, “Qiubai” pear, “Redzaosu” pear, “Shuihongxiao” pear, “Zaomisuan” pear, “Nanguo” pear, “Akiziki” pear, “Yali” pear and “Huagai” pear, with their F′ scores from −0.034 to 0.539. The other eleven pear paste samples, with their F′ scores on the scale of −0.834 to −0.245, clustered together. It was necessary to note that the last group splited into two new clusters at a similarity level of 1, in which “Hongxiangsu” pear, “Xueqing”pear, “Nansui” pear, and “Xinli No. 7” pear were contained in one of the new clusters, just the same as in the bottom four of the lowest F score, which indicated that cluster analysis using F′ scores was accurate.

## 4. Conclusions

It was concluded that the quality of pear paste was closely related to the pear cultivar through the measurement of the physicochemical properties and the antioxidant activities of different pear paste samples. The comprehensive evaluation model for the quality of pear paste was constructed based on only four effective parameters, i.e. *b**, TA, DPPH RSA and FRAP. According to the score of F and F′, pear cultivars of "Mantianhong", "Xiangshui", and "Anli" with higher phenolics and flavonoids content and higher antioxidant activities ranked the top three, conversely, pear cultivars of "Xueqing", "Nansui", "Hongxiangsu", and “Xinli No. 7” with lower phenolics and flavonoids content and lower antioxidant activities ranked the bottom four, and both of them clustered into separate groups respectively, which demonstrating that the model could evaluate the quality of pear paste correctly, and cluster analysis could differentiate the pear paste samples successfully based on calculated score of F′. Therefore, the research shed light on the physicochemical and antioxidant properties of pear paste from different cultivars and established an evaluation method for the quality of pear paste.

## Figures and Tables

**Figure 1 foods-12-00187-f001:**
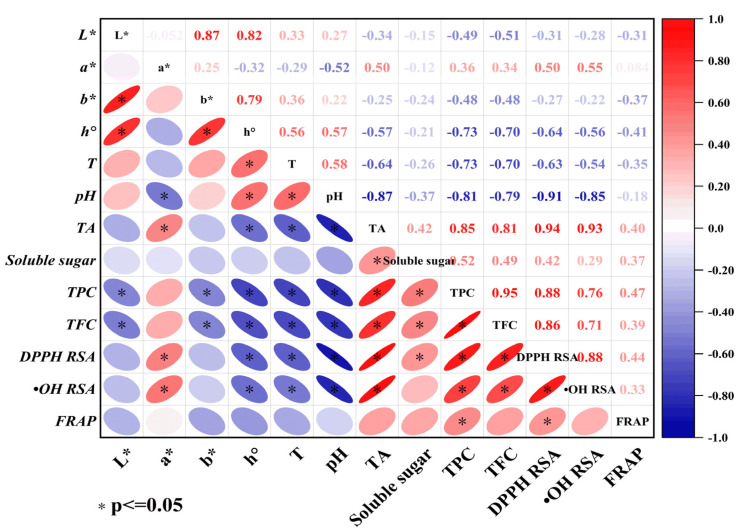
Spearman correlation coefficient heat map of physicochemical properties and antioxidant activities of pear pastes prepared from 23 cultivars. Red color means positive correlation, while blue means negative. Meanwhile, the darker the color and the flatter the ellipse indicates the stronger correlation, while the correlation is not significant when color is light and ellipse is close to a circle. The correlation significant level was set at *p* ≤ 0.05.

**Figure 2 foods-12-00187-f002:**
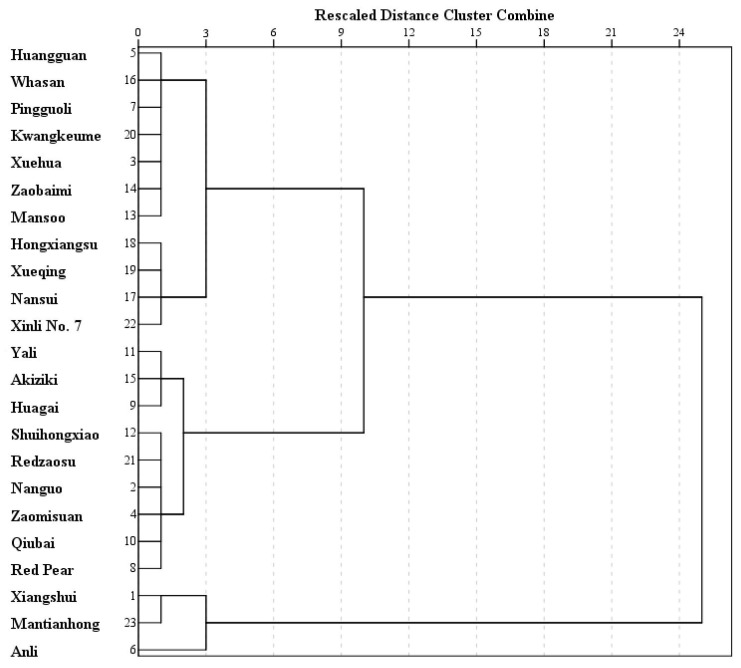
Cluster analysis of pear pastes from 23 cultivars.

**Table 1 foods-12-00187-t001:** Physicochemical properties of different cultivars of pear fruit.

Cultivar	Weight/g	Firmness/N	Juice Yield/%	Pear Paste Yield/%	SSC/°Brix	TA/%	pH
Xiangshui	69.90 ± 21.02	37.83 ± 10.68	69.57	10.72	14.00 ± 0.35	0.60 ± 0.01	3.34 ± 0.02
Nanguo	86.66 ± 9.57	76.05 ± 16.86	73.21	8.92	16.30 ± 0.82	0.48 ± 0.01	3.56 ± 0.01
Xuehua	448.97 ± 109.02	6.54 ± 1.01	85.00	10.64	11.28 ± 0.51	0.10 ± 0.01	4.70 ± 0.01
Zaomisuan	462.68 ± 60.68	64.09 ± 5.68	81.40	10.78	11.10 ± 0.70	0.32 ± 0.01	4.03 ± 0.01
Huangguan	265.94 ± 63.17	53.90 ± 8.53	79.68	10.69	12.67 ± 0.79	0.14 ± 0.01	4.62 ± 0.02
Anli	131.60 ± 42.63	88.00 ± 7.25	84.69	12.06	11.65 ± 0.86	0.79 ± 0.01	3.20 ± 0.01
Pingguoli	311.39 ± 32.38	55.96 ± 6.96	84.05	12.22	12.93 ± 0.68	0.29 ± 0.01	3.89 ± 0.01
Red Pear	200.33 ± 51.77	85.65 ± 15.29	84.69	9.18	10.13 ± 0.73	0.30 ± 0.01	4.10 ± 0.17
Huagai	136.37 ± 17.76	60.86 ± 7.35	83.16	8.92	15.07 ± 0.69	0.49 ± 0.01	3.45 ± 0.02
Qiubai	131.19 ± 16.61	68.11 ± 7.84	84.89	12.00	12.93 ± 0.59	0.24 ± 0.01	4.09 ± 0.03
Yali	307.97 ± 63.06	49.98 ± 4.70	84.79	11.50	12.13 ± 0.64	0.14 ± 0.01	4.66 ± 0.01
Shuihongxiao	283.86 ± 24.63	68.40 ± 8.92	85.04	13.16	12.67 ± 0.49	0.31 ± 0.01	3.84 ± 0.02
Mansoo	292.25 ± 27.62	64.09 ± 9.70	78.91	10.24	11.70 ± 0.44	0.11 ± 0.01	4.92 ± 0.06
Zaobaimi	296.18 ± 29.59	50.57 ± 11.86	83.25	12.45	12.94 ± 0.62	0.15 ± 0.01	4.52 ± 0.35
Akiziki	307.97 ± 31.16	39.89 ± 8.92	84.51	11.19	11.63 ± 0.66	0.13 ± 0.01	4.89 ± 0.01
Whasan	283.86 ± 39.65	58.80 ± 10.88	82.10	12.28	13.70 ± 0.68	0.13 ± 0.01	4.77 ± 0.01
Nansui	255.16 ± 30.76	51.35 ± 6.17	80.08	8.17	13.25 ± 0.50	0.09 ± 0.01	5.38 ± 0.01
Hongxiangsu	258.26 ± 37.51	65.56 ± 5.29	80.69	10.70	10.90 ± 0.61	0.05 ± 0.01	5.45 ± 0.02
Xueqing	483.74 ± 46.21	52.92 ± 9.60	81.04	12.73	13.30 ± 0.44	0.15 ± 0.01	4.77 ± 0.01
Kwangkeume	409.11 ± 65.90	56.94 ± 8.33	81.03	12.53	13.35 ± 0.46	0.22 ± 0.02	4.72 ± 0.01
Redzaosu	346.6 ± 49.66	52.04 ± 5.59	84.15	10.38	11.00 ± 0.59	0.13 ± 0.01	4.80 ± 0.01
Xinli No. 7	239.48 ± 36.10	32.73 ± 4.70	88.09	11.56	10.77 ± 0.55	0.20 ± 0.02	4.43 ± 0.01
Mantianhong	359.21 ± 64.05	63.90 ± 12.15	86.45	10.89	11.35 ± 0.49	0.47 ± 0.01	3.94 ± 0.01
Mean	276.90	56.70	82.19	11.04	12.47	0.26	4.35
SD	114.65	17.33	4.14	1.34	1.47	0.19	0.62
CV/%	41.40	30.56	5.04	12.11	11.81	72.11	14.33

**Table 2 foods-12-00187-t002:** Color parameters and transmittance of pear pastes from 23 cultivars.

Cultivar	*L**	*a**	*b**	*h*⸰	T/%
Xiangshui	43.82 ± 0.14	4.91 ± 0.02	11.87 ± 0.06	67.51 ± 0.01	46.85 ± 1.28
Nanguo	42.50 ± 0.29	2.46 ± 0.13	8.28 ± 0.51	73.44 ± 0.22	51.63 ± 1.70
Xuehua	48.52 ± 0.50	−1.16 ± 0.10	10.68 ± 0.94	96.22 ± 0.13	89.25 ± 1.21
Zaomisuan	45.62 ± 0.97	0.23 ± 0.02	11.76 ± 0.60	88.89 ± 0.14	84.45 ± 1.03
Huangguan	47.25 ± 0.30	−0.58 ± 0.04	11.76 ± 0.60	92.81 ± 0.26	84.10 ± 1.13
Anli	40.30 ± 0.60	4.43 ± 0.11	7.65 ± 0.06	59.94 ± 0.41	31.10 ± 2.69
Pingguoli	44.03 ± 0.13	−0.15 ± 0.05	7.17 ± 0.05	91.17 ± 0.41	70.25 ± 0.49
Red Pear	47.38 ± 0.22	−0.38 ± 0.02	12.34 ± 0.34	91.75 ± 0.11	76.00 ± 1.98
Huagai	43.34 ± 0.06	1.24 ± 0.01	5.53 ± 0.17	77.36 ± 0.35	84.25 ± 1.63
Qiubai	47.98 ± 0.32	−0.80 ± 0.05	12.17 ± 0.08	93.75 ± 0.20	75.80 ± 2.97
Yali	51.81 ± 0.81	−1.29 ± 0.03	14.64 ± 0.34	95.04 ± 0.17	89.10 ± 2.12
Shuihongxiao	47.16 ± 0.16	0.57 ± 0.05	9.09 ± 0.12	86.43 ± 0.34	80.75 ± 0.64
Mansoo	45.88 ± 0.21	0.04 ± 0.05	9.75 ± 0.63	89.78 ± 0.30	87.70 ± 1.84
Zaobaimi	49.87 ± 0.17	−1.19 ± 0.01	10.94 ± 0.25	96.22 ± 0.12	90.45 ± 0.78
Akiziki	47.93 ± 0.11	−0.16 ± 0.04	14.16 ± 0.13	90.65 ± 0.18	88.10 ± 1.98
Whasan	47.84 ± 0.78	−0.74 ± 0.06	10.82 ± 0.14	93.93 ± 0.33	86.95 ± 2.19
Nansui	49.69 ± 0.41	−1.27 ± 0.06	10.10 ± 0.33	97.19 ± 0.09	80.75 ± 1.77
Hongxiangsu	44.53 ± 0.53	0.28 ± 0.05	10.43 ± 0.60	88.45 ± 0.18	58.20 ± 0.42
Xueqing	47.77 ± 0.59	−0.83 ± 0.13	9.03 ± 0.45	95.21 ± 0.56	90.95 ± 0.64
Kwangkeume	45.04 ± 0.34	0.21 ± 0.03	9.87 ± 0.33	88.80 ± 0.17	85.80 ± 0.28
Redzaosu	50.78 ± 0.20	−0.49 ± 0.01	13.95 ± 0.29	92.03 ± 0.07	76.25 ± 0.78
Xinli No. 7	49.12 ± 0.89	−0.78 ± 0.07	8.36 ± 0.35	95.35 ± 0.22	86.70 ± 0.28
Mantianhong	40.63 ± 0.14	4.80 ± 0.13	7.96 ± 0.05	58.91 ± 0.53	44.00 ± 1.50
Mean	46.47	0.41	10.36	87.00	75.63
SD	3.07	1.91	2.32	11.43	17.15
CV/%	6.61	470.79	22.42	13.14	22.68

**Table 3 foods-12-00187-t003:** Physicochemical properties of pear pastes from 23 cultivars.

Cultivar	pH	TA%	Soluble Sugar/(mg/mL)	TPC / (mg/100 mL)	TFC / (mg/100 mL)
Xiangshui	3.40 ± 0.10	0.372 ± 0.005	78.86 ± 3.39	43.22 ± 1.59	20.44 ± 0.87
Nanguo	3.64 ± 0.10	0.282 ± 0.002	76.06 ± 3.94	29.94 ± 2.63	17.42 ± 1.93
Xuehua	4.76 ± 0.17	0.108 ± 0.003	78.51 ± 4.29	14.78 ± 0.52	4.44 ± 0.20
Zaomisuan	4.12 ± 0.13	0.289 ± 0.009	67.28 ± 1.61	18.94 ± 2.54	7.26 ± 0.29
Huangguan	4.73 ± 0.08	0.155 ± 0.032	71.00 ± 5.75	13.55 ± 0.22	2.80 ± 0.06
Anli	3.27 ± 0.09	0.652 ± 0.013	77.33 ± 2.6	38.15 ± 0.91	16.04 ± 0.93
Pingguoli	3.99 ± 0.21	0.233 ± 0.002	80.60 ± 1.99	15.65 ± 0.81	8.04 ± 0.60
Red Pear	4.17 ± 0.07	0.331 ± 0.009	77.79 ± 2.76	26.71 ± 0.36	10.20 ± 0.10
Huagai	3.54 ± 0.11	0.336 ± 0.002	77.38 ± 2.41	25.85 ± 0.44	13.61 ± 0.16
Qiubai	4.13 ± 0.10	0.208 ± 0.002	73.91 ± 2.82	20.98 ± 0.64	9.12 ± 0.25
Yali	4.75 ± 0.03	0.135 ± 0.002	63.04 ± 2.49	11.53 ± 0.35	3.32 ± 0.03
Shuihongxiao	3.91 ± 0.11	0.231 ± 0.007	68.66 ± 1.95	30.10 ± 0.66	14.17 ± 1.36
Mansoo	4.98 ± 0.08	0.126 ± 0.005	79.53 ± 3.16	16.38 ± 0.12	5.33 ± 0.13
Zaobaimi	4.80 ± 0.09	0.126 ± 0.004	68.66 ± 3.16	11.85 ± 0.43	2.83 ± 0.19
Akiziki	4.95 ± 0.10	0.158 ± 0.004	72.18 ± 1.53	15.32 ± 0.74	3.51 ± 0.17
Whasan	4.84 ± 0.11	0.166 ± 0.002	84.63 ± 2.78	12.79 ± 2.04	3.68 ± 0.24
Nansui	5.36 ± 0.10	0.088 ± 0.002	59.47 ± 3.91	10.52 ± 0.98	2.74 ± 0.04
Hongxiangsu	5.31 ± 0.08	0.068 ± 0.006	68.30 ± 1.27	13.60 ± 0.61	3.86 ± 0.15
Xueqing	4.86 ± 0.11	0.111 ± 0.006	67.08 ± 0.84	11.30 ± 0.67	3.13 ± 0.18
Kwangkeume	4.82 ± 0.09	0.161 ± 0.006	70.60 ± 3.36	12.37 ± 0.33	3.13 ± 0.04
Redzaosu	4.52 ± 0.11	0.179 ± 0.004	74.93 ± 1.25	18.29 ± 0.92	6.59 ± 0.93
Xinli No. 7	4.88 ± 0.11	0.135 ± 0.084	67.89 ± 4.72	12.51 ± 0.12	3.85 ± 0.13
Mantianhong	4.02 ± 0.08	0.397 ± 0.004	71.62 ± 2.30	51.98 ± 1.37	42.28 ± 0.92
Mean	4.42	0.219	72.84	21.90	9.03
SD	0.61	0.133	6.11	11.20	8.79
CV/%	13.69	60.605	8.39	53.65	99.64

**Table 4 foods-12-00187-t004:** Antioxidant activities of pear pastes from 23 cultivars.

Cultivar	DPPH RSA/%	^•^OH RSA/%	FRAP/OD_700_
Xiangshui	78.08 ± 0.78	83.84 ± 3.05	0.48 ± 0.02
Nanguo	61.77 ± 1.48	82.49 ± 3.13	0.26 ± 0.02
Xuehua	14.73 ± 1.35	65.45 ± 1.02	0.05 ± 0.01
Zaomisuan	38.96 ± 1.70	92.20 ± 0.65	0.11 ± 0.01
Huangguan	15.07 ± 0.96	72.84 ± 1.31	0.14 ± 0.03
Anli	65.84 ± 0.46	96.16 ± 0.25	0.23 ± 0.04
Pingguoli	34.03 ±2.55	82.96 ± 1.75	0.21 ± 0.01
Red Pear	44.73 ± 0.49	86.53 ± 0.70	0.22 ± 0.04
Huagai	53.47 ± 0.98	86.93 ± 1.79	0.41 ± 0.02
Qiubai	47.43 ± 0.52	78.63 ± 0.23	0.23 ± 0.02
Yali	16.99 ± 2.05	74.65 ± 3.42	0.10 ± 0.02
Shuihongxiao	62.11 ± 1.02	81.81 ± 1.06	0.31 ± 0.01
Mansoo	12.89 ± 1.03	71.23 ± 0.66	0.50 ± 0.02
Zaobaimi	7.58 ± 0.53	74.15 ± 3.19	0.22 ± 0.03
Akiziki	16.91 ± 0.78	73.29 ± 0.69	0.37 ± 0.07
Whasan	17.81 ± 1.45	70.55 ± 2.63	0.29 ± 0.10
Nansui	1.50 ± 0.22	62.49 ± 0.93	0.36 ± 0.08
Hongxiangsu	2.53 ± 0.46	57.02 ± 3.07	0.18 ± 0.04
Xueqing	10.02 ± 1.29	68.16 ± 1.06	0.09 ± 0.01
Kwangkeume	18.79 ± 0.15	82.14 ± 0.89	0.25 ± 0.03
Redzaosu	31.51 ± 0.86	78.14 ± 1.60	0.26 ± 0.02
Xinli No. 7	6.34 ± 0.53	59.72 ± 1.95	0.04 ± 0.01
Mantianhong	69.31 ± 1.05	92.81 ± 0.26	2.05 ± 0.06
Mean	31.67	77.14	0.32
SD	24.01	10.54	0.40
CV/%	75.81	13.66	124.26

**Table 5 foods-12-00187-t005:** Factor matrix including loadings, eigenvalues, percent of variance and cumulative variance for the first four principal factors.

Variable	F1	F2	F3	F4
*L**	−0.508	−0.405	0.662	−0.181
*a**	0.669	0.582	−0.347	0.058
*b**	−0.096	−0.096	0.928	−0.084
*h*⸰	−0.679	−0.576	0.385	−0.044
T	−0.610	−0.482	0.353	0.009
pH	−0.915	−0.094	0.217	−0.217
TA	0.881	0.227	−0.253	0.119
Soluble sugar	0.233	0.036	−0.126	0.958
TPC	0.714	0.640	−0.176	0.121
TFC	0.527	0.795	−0.197	0.047
DPPH RSA	0.871	0.355	−0.094	0.176
FRAP	0.093	0.957	−0.075	0.021
^•^OH RSA	0.862	0.199	−0.016	0.134
Eigenvalue	5.516	3.260	1.903	1.090
% Variance	42.4	25.1	14.6	8.4
% Cumulative	42.4	67.5	82.1	90.5
Principal factor weight	0.469	0.277	0.162	0.093

**Table 6 foods-12-00187-t006:** Factor matrix including loadings, eigenvalues, percent of variance and cumulative variance for the first four principal factors.

Cultivar	F_1_	F_2_	F_3_	F_4_	F	Rank	F′	Rank
Mantianhong	0.216	4.184	−0.338	−0.366	1.171	1	1.154	1
Xiangshui	1.630	0.895	0.537	0.544	1.150	2	1.134	2
Anli	2.456	−0.476	−1.553	−0.265	0.744	3	0.777	3
Red Pear	0.814	−0.317	1.178	0.836	0.563	4	0.539	4
Redzaosu	0.149	0.017	1.789	0.447	0.406	5	0.324	6
Qiubai	0.437	−0.280	1.009	0.291	0.318	6	0.391	5
Zaomisuan	1.135	−0.859	0.567	−0.982	0.295	7	0.287	8
Nanguo	0.928	0.039	−1.056	0.146	0.289	8	0.276	9
Shuihongxiao	0.759	−0.128	0.070	−0.698	0.267	9	0.321	7
Yali	0.108	−0.418	1.828	−1.473	0.094	10	0.066	11
Akiziki	−0.460	0.297	1.314	0.142	0.093	11	0.140	10
Huagai	0.636	−0.470	−1.557	0.645	−0.024	12	−0.034	12
Whasan	−1.011	−0.046	0.286	2.236	−0.233	13	−0.245	13
Huangguan	−0.371	−0.386	0.297	−0.171	−0.249	14	−0.247	14
Pingguoli	0.038	−0.687	−1.192	1.255	−0.249	15	−0.354	15
Kwangkeume	−0.305	−0.385	−0.447	−0.298	−0.350	16	−0.362	16
Zaobaimi	−0.569	−0.396	0.409	−0.496	−0.356	17	−0.468	18
Mansoo	−1.261	0.460	−0.323	1.400	−0.386	18	−0.424	17
Xuehua	−0.942	−0.367	0.130	1.207	−0.410	19	−0.477	19
Xueqing	−0.798	−0.587	−0.670	−0.800	−0.720	20	−0.716	22
Nansui	−1.209	0.121	−0.265	−2.064	−0.769	21	−0.658	20
Hongxiangsu	−1.354	0.338	−1.095	−0.803	−0.793	22	−0.697	21
Xinli No. 7	−1.025	−0.550	−0.919	−0.733	−0.850	23	−0.834	23

## Data Availability

The data presented in this study are available on request from the corresponding author.

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
