# Peer review of "Chinese Traditional Pear Paste: Physicochemical Properties, Antioxidant Activities and Quality Evaluation"

_foods, 2023, doi:10.3390/foods12010187_

Round 1

Reviewer 1 Report

The article is original since there are no reports of similar works in the literature according to the authors. The article is a valuable and novel contribution to the topics related to the knowledge and understanding of the physicochemical and potential functional properties of pear paste. It is well-written and presented in a logical sequence. The background is in agreement with the investigation. The objective is clear and precise and the methodology is consistent with the development of the research. Each topic ends with a clear conclusion that allows an easy understanding of the detail in the explanation. Tables are presented that summarizes the most important information, which also helps to make the content easier to understand. Also, a detailed statistical analysis of the data has been carried out.

The authors should consider the following minor changes

Methodology

-Line 135: For Folin Ciocalteau method the samples were incubated during 2 hours before measurement in spectrophotometer?. Samples are often incubated for 30 min.

- unit of expression of results of FRAP

Results

-Line 188: The single pear fruit weight, firmness, SSC, TA and pH of pear juice, juice and pear 188 paste yields. Is pear juice o fruit?

-LINE 221: authors should explain what could be attributed to the difference in TPC and TFC

Line 290: eliminate the word “the”. It is repeated

Reviewer 2 Report

Review on manuscript: foods-2094019

Chinese Traditional Pear Paste: Physicochemical Properties, Antioxidant Activities and Quality Evaluation

by Yunxiao Feng, Hong Cheng, Yudou Cheng, Jiangli Zhao, Jingang He, Nan Li, Jinxiao Wang and Jun-feng Guan

submitted to Foods

In the manuscript submitted for comments, the authors evaluated and differentiated the quality of pear pastes prepared from different cultivars based on their physicochemical properties and antioxidant activities. In my opinion the manuscript is prepared correctly with minor changes.

General commend: spaces are missing from the text in many places, e.g. lines 33, 35, 37 ...

Detailed recommendation

lines 13-16 – the specific results of the analyzes carried out should be given,

lines 19-25 – this description is too general and it is not entirely clear what it means,

lines 73-86 – origin country should be added,

line 117 – type of illuminant and measuring geometry should be given,

line 137 – model, producer and origin country of spectrophotometer should be given,

lines 164 and 176 – instead of rpm the centrifugal force should be specified,

line 167 – subscript should be used,

line 173 – abbreviation should be explained,

Discussion - the discussion lacks references to literature, and the authors too often provide data contained in tables in the text,

Table 1 – firmness should be given in newtons,

Tale 4 – unit is missing for FRAP value,

lines 260-263 – are the reported values of correlation coefficients statistically significant?

Reviewer 3 Report

The work is at good scientific level, however, it requires editorial corrections and the development of the discussion in the light of the results obtained by other authors.

Put space between word and bracket, for example line 40 vinegar[18−20]. Please correct in whole document

Line 167 change FeSO4 to FeSO4

 Regarding the Antioxidant Potential Analysis (DPPH): to compare the obtained results with literature data, it is more convenient to use the IC50 parameter.

The obtained results should be discussed to a greater extent (if possible) in the light of the available literature data (if possible). The authors refer very little to the results obtained by other authors.
